# Diagnostic Accuracy of Adrenal Iodine-131 6-Beta-Iodomethyl-19-Norcholesterol Scintigraphy for the Subtyping of Primary Aldosteronism

**DOI:** 10.3390/biomedicines11071934

**Published:** 2023-07-07

**Authors:** Marta Araujo-Castro, Miguel Paja Fano, Marga González Boillos, Eider Pascual-Corrales, Ana María García Cano, Paola Parra Ramírez, Patricia Martín Rojas-Marcos, Almudena Vicente Delgado, Anna Casteràs, Albert Puig, Iñigo García Sanz, Patricia Díaz Guardiola, Cristina Robles Lázaro, Miguel Antonio Sampedro Núñez, Raquel Guerrero-Vázquez, María del Castillo Tous, Theodora Michalopoulou Alevras, Susana Tenes Rodrigo, Felicia A. Hanzu

**Affiliations:** 1Endocrinology & Nutrition Department, Hospital Universitario Ramón y Cajal, 28034 Madrid, Spain; 2Instituto de Investigación Biomédica Ramón y Cajal (IRYCIS), 28034 Madrid, Spain; 3Medicine Department, University of Alcalá, 28801 Madrid, Spain; 4Endocrinology & Nutrition Department, OSI Bilbao-Basurto, Hospital Universitario de Basurto, 48013 Bilbao, Spain; 5Medicine Department, Basque Country University, 48013 Bilbao, Spain; 6Endocrinology & Nutrition Department, Hospital Universitario de Castellón, 12004 Castellón, Spain; 7Biochemistry Department, Hospital Universitario Ramón y Cajal, 28034 Madrid, Spain; 8Endocrinology & Nutrition Department, Hospital Universitario La Paz, 28046 Madrid, Spain; 9Endocrinology & Nutrition Department, Hospital Universitario de Toledo, 45007 Toledo, Spain; 10Endocrinology & Nutrition Department, Hospital Universitario de Vall Hebron, 08035 Barcelona, Spain; 11General & Digestive Surgery Department, Hospital Universitario de La Princesa, 28006 Madrid, Spain; 12Endocrinology & Nutrition Department, Hospital Universitario Infanta Sofía, 28702 Madrid, Spain; 13Endocrinology & Nutrition Department, Complejo Universitario de Salamanca, 37007 Salamanca, Spain; 14Endocrinology & Nutrition Department, Hospital Universitario La Princesa, 28006 Madrid, Spain; 15Endocrinology & Nutrition Department, Hospital Virgen de la Macarena, 41009 Sevilla, Spain; 16Endocrinology & Nutrition Department, Hospital Joan XXIII, 43005 Tarragona, Spain; 17Internal Medicine, Hospital Infanta Leonor de Vallecas, 28031 Madrid, Spain; 18Endocrinology & Nutrition Department, Hospital Clinic, IDIPAS, 08007 Barcelona, Spain

**Keywords:** primary aldosteronism, adrenal venous sampling, norcholesterol adrenal scintigraphy, lateralization index, adrenalectomy

## Abstract

Purpose: To evaluate the diagnostic accuracy of the 131I-6β-iodomethyl-19-norcholesterol (NP-59) adrenal scintigraphy for the subtyping diagnosis of primary aldosteronism (PA), considering as gold standard for the diagnosis of unilateral PA (UPA), either the results of the adrenal venous sampling (AVS) or the outcome after adrenalectomy. Methods: A retrospective multicenter study was performed on PA patients from 14 Spanish tertiary hospitals who underwent NP-59 scintigraphy with an available subtyping diagnosis. Patients were classified as UPA if biochemical cure was achieved after adrenalectomy or/and if an AVS lateralization index > 4 with ACTH stimulation or >2 without ACTH stimulation was observed. Patients were classified as having bilateral PA (BPA) if the AVS lateralization index was ≤4 with ACTH or ≤2 without ACTH stimulation or if there was evidence of bilateral adrenal nodules >1 cm in each adrenal gland detected by CT/MRI. Results: A total of 86 patients with PA were included (70.9% (n = 61) with UPA and 29.1% (n = 25) with BPA). Based on the NP-59 scintigraphy results, 16 patients showed normal suppressed adrenal gland uptake, and in the other 70 cases, PA was considered unilateral in 49 patients (70%) and bilateral in 21 (30%). Based on 59-scintigraphy results, 10.4% of the patients with unilateral uptake had BPA, and 27.3% of the cases with bilateral uptake had UPA. The AUC of the ROC curve of the NP-59 scintigraphy for PA subtyping was 0.812 [0.707–0.916]. Based on the results of the CT/MRI and NP-59 scintigraphy, only 6.7% of the patients with unilateral uptake had BPA, and 24% of the cases with bilateral uptake had UPA. The AUC of the ROC curve of the model combining CT/MRI and 59-scintigraphy results for subtyping PA was 0.869 [0.782–0.957]. Conclusion: The results of NP-59 scintigraphy in association with the information provided by the CT/MRI may be useful for PA subtyping. However, their diagnostic accuracy is only moderate. Therefore, it should be considered a second-line diagnostic tool when AVS is not an option.

## 1. Introduction

Primary aldosteronism (PA) is the most common cause of endocrine and secondary hypertension, being present in 20% of all cases of resistant hypertension and in 5–10% of the total hypertensive population [1,2]. Moreover, patients with PA have been found to have a higher risk of cardiovascular events, such as myocardial infarction, coronary artery disease, stroke, and atrial fibrillation, than patients with essential hypertension matched by blood pressure levels [3]. The specific treatment with surgery or, if not feasible, with effective mineralocorticoid receptor blockade reduces the cardiometabolic risk of these patients [4,5]. The distinction between the cases of unilateral aldosterone-producing adenoma and bilateral adrenal hyperplasia is challenging but crucial for the decision on further treatment: medication in the case of bilateral adrenal hyperplasia and surgery in the case of unilateral aldosterone-producing adenoma. Adrenal venous sampling (AVS) is a decisive method for the lateralization of PA [2,6]. However, AVS is a technically challenging procedure, and the rate of catheterization failure is frequently high, especially in the right adrenal vein, due to the anatomical characteristics of this vascular region [7,8]. On the other hand, although CT or MRI are currently the default imaging tools to localize aldosterone-producing adenomas, their diagnostic performance is quite low, according to some studies [9].

It has been proposed that if AVS fails, 131 I-6-β-iodomethyl-19-norcholesterol (NP-59) scintigraphy can be used to determine the subtype of PA [10]. Adrenal scintigraphy with the NP-59 radiotracer can visualize the adrenal cortex and assess its hormonal activity. In addition, it is a minimally invasive procedure and has a high success rate, regardless of the operator’s skill [11,12]. Moreover, the administration of dexamethasone, which suppresses adrenal cortical hormones, markedly improves the diagnostic usefulness of adrenal scintigraphy [12].

Considering this background, our study aimed to assess the diagnostic accuracy of NP-59-adrenal scintigraphy results in subtyping PA, considering the gold standard for diagnosing UPA as either the AVS results or the biochemical cure achieved through adrenalectomy.

## 2. Materials and Methods

### 2.1. Study Population and Definitions

Patients with PA in follow-up between January 2018 and December 2022 were enrolled in 30 Spanish tertiary hospitals (SPAIN-ALDO registry). When the data were analyzed (7 April 2023), a total of 833 patients had been included in the database. The inclusion criteria to enter this specific study were the following: (i) available results of a NP-59 adrenal scintigraphy for the subtyping diagnosis of PA and demonstration of complete biochemical cure after adrenalectomy; or (ii) available results of a NP-59 adrenal scintigraphy for the subtyping diagnosis of PA and available AVS demonstrating unilateral or bilateral PA or bilateral adrenal nodules >1 cm in CT/MRI. A total of 125 patients underwent adrenal scintigraphy, of whom 86 met inclusion criteria and had been evaluated in 14 centers. These patients were classified in two groups: (i) UPA if biochemical cure was achieved (based on the PASO classification system [13]) after adrenalectomy or/and if the lateralization index in AVS > 4 with ACTH or > 2 without ACTH stimulation (n = 61); and (ii) Bilateral PA (BPA) if lateralization index in AVS was ≤ 4 with ACTH or ≤ 2 without ACTH stimulation or if there was evidence of bilateral adrenal nodules >1cm in the CT/MRI (n = 25) (Figure 1).

As we have previously described [14], the clinical data of the patients were entered into an electronic database (REDCap^®^ database) [15,16] after pseudonymization using an identification number (record_Id). The SPAIN-ALDO registry includes data on demographic characteristics, comorbidities, biochemical, and radiological parameters, as well as information on physical evaluation and treatments for PA, as we have previously mentioned [17]. The study was approved by the Ethics committee of the Ramón y Cajal Hospital, Madrid.

### 2.2. Dexamethasone-Suppression NP-59 Scintigraphy

To suppress the normal adrenal uptake of NP-59, different protocols were used across the different centers: (i) the patients took oral dexamethasone 2 mg daily (1.0 mg every 12 h) 3 days before and 7 days after the intravenous injection of NP-59 (n = 58); (ii) some centers used a higher dose of dexamethasone (4 mg daily, using 1 mg every 6 h) 7 days before and 2 days after the intravenous injection of NP-59 (n = 12). Saturated potassium iodide (50 mg/d) was also administered orally for blockade of thyroidal uptake for the same duration as that for oral dexamethasone administration. In addition, some drugs are recommended to be suspended before the scintigraphy (spironolactone and other diuretics; captopril; calcium antagonists; oral hormonal contraceptives; and cholestyramine). The following variables were registered: (i) adrenal NP-59 uptake or not; and (ii) unilateral or bilateral hyper uptake.

### 2.3. Statistical Analysis

All statistical analyses were conducted with STATA.15. Shapiro–Wilk’s test was used to assess the normality of continuous variables. All data are expressed as the mean and SD for normally distributed variables and the median (25th–75th percentile) for non-normally distributed variables. The Student’s *t*-test was used to compare quantitative variables and the *X*^2^ test for qualitative variables between two groups. The lineal correlation between continuous parameters was determined by Pearson’s correlation coefficient (r). Reliability was evaluated with the kappa index and the specific positive and negative agreement indexes. Nonparametric receiver-operator curve (ROC) analysis was used to determine the diagnostic accuracy of the results of the NP-59 scintigraphy for the diagnosis of UPA (vs. BPA). In all cases, a two-tailed *p* value < 0.05 was considered statistically significant.

## 3. Results

### 3.1. Baseline Characteristics

A total of 86 patients with PA, available NP-59 scintigraphy, and a confirmed diagnosis of UPA or BPA were included. Most of them were classified as UPA (70.9%, n = 61), and the remaining 29.1% (n = 25) as BPA. No statistically significant differences were detected in baseline characteristics between both groups, except for a higher prevalence of unilaterality in UPA based on CT/MRI results compared to those with BPA (OR 20.2, 95% CI 5.38–75.96). Differences in baseline characteristics between both groups are shown in Table 1. The prevalence of autonomous cortisol secretion based on the 1.8 µg/dL threshold in the dexamethasone suppression test was 50% (n = 15/30).

### 3.2. Diagnostic Accuracy of NP-59 Scintigraphy to Differentiate Unilateral and Bilateral PA

The main reasons for performing NP-59 scintigraphy were inconclusive AVS in 16 cases, low rate of proper catheterization in the AVS in the center (n = 9), bilaterality according to CT/MRI (n = 8), AVS not available in the center (n = 6), and refusal to perform the AVS by the patient (n = 4). In the other 43 cases, NP-59 scintigraphy was performed instead of the AVS for PA subtyping. Based on the NP-59 scintigraphy results, no evidence of uptake was demonstrated in 16 patients. In the other 70 cases, PA was considered unilateral in 49 patients (70%) and bilateral in 21 patients (30%). Out of the 16 patients who underwent scintigraphy without evidence of NP-59 uptake, 93.3% had adrenal lesions smaller than 2 cm based on CT/MRI results. The median tumor size of these 16 cases was 12.9 mm (range 0 to 30). Most of these patients had a UPA (81.3%, n = 13), and three had a BPA.

The degree of agreement (reliability) between the results of the NP-59 scintigraphy and our definition for differentiating UPA and BPA was moderate, with a kappa index of 0.631. The positive specific agreement (to classify unilateral disease) was 88.7% [80.8% to 93.5%], the negative specific agreement (to classify bilateral disease) was 74.4% [59.8% to 74.4%], with a global agreement of 84.3% [74.0% to 91.0%]. Based on NP-59 scintigraphy results, 10.4% of the patients with unilateral hyper uptake had BPA, and 27.3% of the cases with bilateral hyper uptake had UPA (Table 2). The results of the NP-59 scintigraphy to differentiate between UPA and BPA yielded an area under the ROC curve (AUC) of 0.812 [0.707–0.916] (Figure 2). The degree of agreement between the results of the NP-59 scintigraphy and the definition that we have used for differentiating UPA and BPA increased to 0.712 when the results of the CT/MRI were included, with a positive specific agreement of 90.3% and a negative specific agreement of 80.9%. In this context, only 6.7% of the patients with unilateral hyper uptake had BPA, and 24% of the cases with bilateral hyper uptake had UPA. Thus, 93.3% of the patients with unilateral disease based on both CT/MRI and NP-59 scintigraphy had UPA. Patients with bilateral disease according to CT/MRI and NP-59 scintigraphy were properly classified in 76% of the cases (Table 3). The AUC of the CT/MRI associated with NP-59 scintigraphy for subtyping PA was 0.869 [0.782–0.957] (Figure 2). In Table 4, we describe the clinical and radiological data of those PA cases wrongly classified as unilateral or bilateral PA according to the NP-59 scintigraphy results compared to those cases properly classified (Table 4). When we compared patients with UPA classified as bilateral PA according to NP-59 scintigraphy and those properly classified as unilateral PA, patients properly classified had unilateral adrenal lesions on CT/MRI more commonly than the former (93.3% vs. 40.9%, *p* < 0.001). However, no differences in tumor size were observed between both groups (20.6 ± 7.80 vs. 18 ± 4.24 mm, *p* = 0.290). On the other hand, patients with bilateral PA with congruent NP-59 scintigraphy results had smaller adrenal lesions than those with unilateral uptake (17.5 ± 9.90 vs. 27.6 ± 6.05 mm, *p* = 0.040) and had unilateral adrenal lesions in CT/MRI less commonly (6.7% vs. 59.1%, *p* < 0.001).

When we only included in the analysis, those patients with unilateral PA according to CT/MRI (unilateral adrenal lesion > 1 cm and normal contralateral adrenal gland, n = 65), there were 13 cases without uptake in the NP-59 scintigraphy. In the 52 cases in which hyper uptake was demonstrated, 86.5% (n = 45) had unilateral hyper and 13.5% (n = 7) had bilateral hyper.

Surgery was performed on 58 patients with UPA. A total of 45 patients were operated based on NP-59 scintigraphy, 3 based on AVS results, and 10 based on CT/MRI results.

## 4. Discussion

PA is considered one of the main causes of secondary hypertension. Adrenalectomy is feasible for unilateral diseases. However, surgical procedures should generally be avoided in BPA, and this condition should be managed with targeted medical therapy [18]. Therefore, the decision to lateralize is critical to improving PA management. For this purpose, AVS is considered the gold standard [2]. However, several limitations, including the fact that it is a highly technical procedure, have limited its clinical application. In this way, the dexamethasone-suppression NP-59 scintigraphy has been proposed as a useful technique for distinguishing UPA and BPA [10,19]. NP-59 is a radioiodine-labeled cholesterol analog with radioisotopic activity. 131I is an appropriate radiotracer for adrenal cortical imaging due to its greater affinity for the adrenal cortex. Dexamethasone given before the study to suppress ACTH enhanced the functional difference between zonae glomerulosa and fasciculata, leading to a better differentiation of PA lateralization. In normal conditions, no focal tracer uptake will be seen prior to the fifth day of tracer injection. Early visualization (before day 5) indicates UPA or BPA [20].

The rate of UPA in our study was 71%, which is quite higher than the rate described in other series [21]. However, there are other studies reporting similar rates of UPA [22]. For example, the Kim SH study [22] found that the prevalence of UPA was 70.7% (aldosterone-producing adenoma or unilateral adrenal hyperplasia), while only 29.3% of the patients had bilateral disease. Differences across studies may be related to the inclusion of different study populations in these studies since, for example, it is reported that the rate of UPA is lower in Japanese than in European centers [23]. The prevalence of UPA is also higher in young patients [24]. In addition, we are aware that a potential selection bias may have occurred in our study since those patients with more chances of having a unilateral PA were more commonly submitted to the AVS. In addition, the rate of hypokalemia and severe hypertension reached prevalences above 60%. Nevertheless, despite this observation, we did not find differences in PAC levels between unilateral and bilateral cases of PA. This finding may be related to the fact that patients with high suspicion of bilateral disease based on CT/MRI and mild PA were less likely to be submitted to AVS and NP-59 scintigraphy and were commonly medically treated without further investigations.

We found that in 50% of the patients, NP-59 scintigraphy was used instead of AVS for PA subtyping. In the other half of the cases, AVS was inconclusive or other challenges related to the procedure led to the election of NP-59 scintigraphy for subtyping. In relation to the high proportion of patients who underwent NP-59 adrenal scintigraphy as the first procedure for PA subtyping, it is important to highlight that NP-59 adrenal scintigraphy is applied for PA lateralization in many Asian and some European countries [25]. For example, in the Japan Endocrine Society guidelines for PA management, dexamethasone-suppression 131I-adosterol scintigraphy is included in the imaging studies section as a valid procedure [25]. In addition, the Taiwan Society of Aldosteronism suggests NP-59 adrenal scintigraphy for functioning tumor lateralization for lesions larger than 1 cm [26]. Nevertheless, in most European countries and in EEUU, AVS is still considered the first-line technique for PA subtyping. Another point to consider when NP-59 scintigraphy results are evaluated is the potential existence of bias in the use of the NP-59 in selecting cases, such as limiting it to cases where it is clinically difficult to determine whether the disease is unilateral or not. Thus, representing the most challenging cases for PA subtyping.

More than 22% of the patients did not have evidence of uptake in the NP-59 scintigraphy, and more than 90% of them had an adrenal lesion in the CT/MRI smaller than 20 mm in diameter, with a median tumor size of 13 mm. It is well known that due to the resolution limit of NP-59 planar imaging, lesions smaller than 15 mm in diameter may not be visible on planar scintigraphy [2]. Supporting this finding, Kloos RT. et al. [27] observed that the number of nonlateralizing images with NP-59 scintigraphy increased from 0% in adrenal lesions > 2 cm to 11% in lesions between 1 and 2 cm and to 48% in adrenal lesions ≤ 1 cm. These observations suggest that NP-59 scintigraphy should be reserved for the study of adrenal lesions larger than 1–2 cm.

We found that the AUC in the ROC curve of the NP-59 scintigraphy to differentiate UPA versus BPA was 0.812 but increased to 0.869 when the 59-scintigraphy results was combined with the information from the CT/MRI. Similar results have been previously described in the Ming-Hsien Wu study [19]. They found that the sensitivity and positive predictive value of NP-59 scintigraphy for UPA detection were 83.3% and 92.3%, respectively, and increased to 85.0% and 89.5% when single-photon emission CT/CT was used in combination with NP-59 scintigraphy. Nevertheless, other studies reported lower degrees of accuracy [28]. For example, the accuracy of NP-59 scintigraphy was 71% in PA in the Kazerooni EA. et al. series [28]. However, the diagnostic accuracy increased in the last available series, reaching a diagnostic accuracy of 83.3% [19], a sensitivity of 90.9%, and a positive predictive value of 83.3%, according to other authors [29]. This improvement is probably related to the introduction of the simultaneous use of integrated SPECT/CT to provide both functional and anatomical information [12]. Thus, considering its high diagnostic accuracy and the non-invasive nature of the test, we recommend that NP-59 scintigraphy should be highly considered in patients with allergies, chronic kidney diseases, and when AVS is not an option. Nevertheless, some limitations of the NP-59 scintigraphy should be taking into account, such as the requirement for prolonged dexamethasone suppression (with the secondary side effects of glucocorticoid treatment), sequential imaging, and limited tracer availability [30].

In our series, based on 59-scintigraphy results, 10.4% of the patients with unilateral uptake had BPA, and 27.3% of the cases with bilateral uptake had UPA. In contrast with our results, a bilateral symmetrical adrenal uptake of radioactive tracer usually suggests BPA [31]. However, bilateral asymmetric uptake of NP-59 provides limited ability to distinguish adenoma from partially suppressed normal contralateral adrenal glands or bilateral hyperplasia. For this reason, some authors have proposed that the degree of adrenal uptake asymmetry could be used to discriminate adenoma from hyperplasia [31]. In the same vein, Weinberger et al. [32] reported a misdiagnosis of UPA as bilateral when there is accumulation on the contralateral side. The use of quantitative analysis may improve the diagnostic accuracy of separating UPA and BPA [33,34]. In addition, other radiotracers, such as 11C-metomidate, may significantly increase the diagnostic capacity to differentiate unilateral and bilateral PA [35]. Burton et al. showed a ratio of tumor maximum standardized uptake values (SUVmax) to normal background adrenal glands greater than 1.25, and absolute tumor SUVmax > 17 reached a specificity of 100% for PA subtyping [35]. The main limitation with the 11C-metomidate PET-CT is that 11C-isotopes have a 20-min half-life, requiring an on-site cyclotron [30], which is not readily available in most centers in Spain.

We are aware that our study has some limitations. First, our PA patients were retrospectively collected from the patient lists of individuals who underwent NP-59 adrenal scintigraphy at some point. Second, due to the absence of successful AVS in all patients, the accuracy of AVS and NP-59 scintigraphy could not be compared precisely. Nevertheless, this is one of the largest studies evaluating the diagnostic accuracy of the results of the NP-59 adrenal scintigraphy for the subtyping diagnosis of PA in the Spanish population.

## 5. Conclusions

The results of NP-59 scintigraphy in association with the information provided by the CT/MRI may be useful for PA subtyping. Their diagnostic accuracy is above 85%, so it should be considered a second-line diagnostic tool when AVS is unfeasible or in patients with allergies to iodine intravenous contrast agents or with severe chronic renal disease. However, the accuracy of NP-59 alone seems to be less reliable for determining PA lateralization.

## Figures and Tables

**Figure 1 biomedicines-11-01934-f001:**
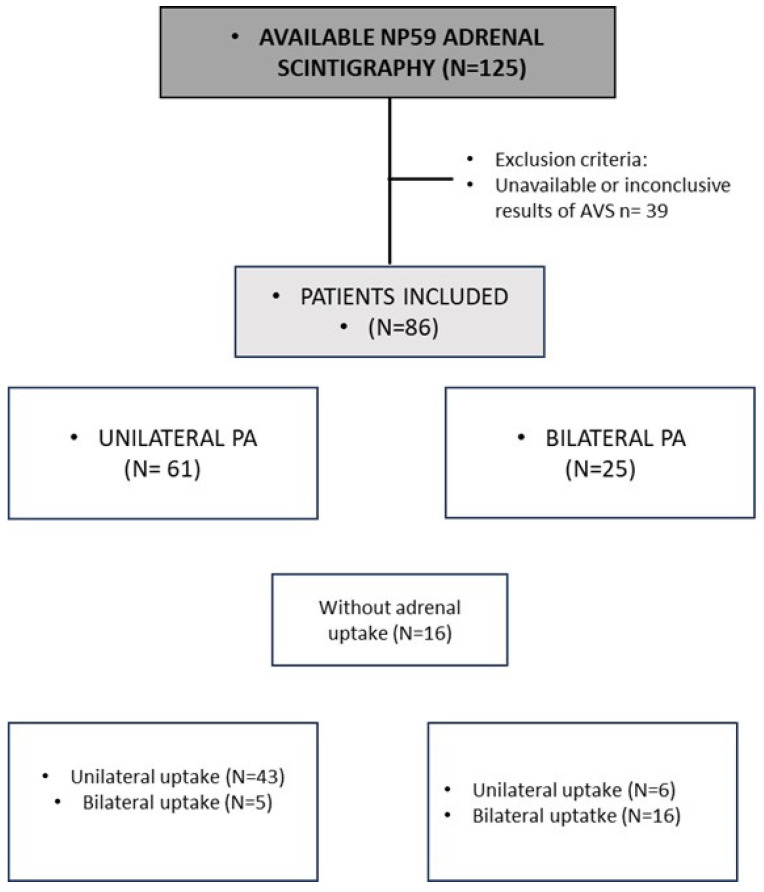
Study population. PA: primary aldosteronism.

**Figure 2 biomedicines-11-01934-f002:**
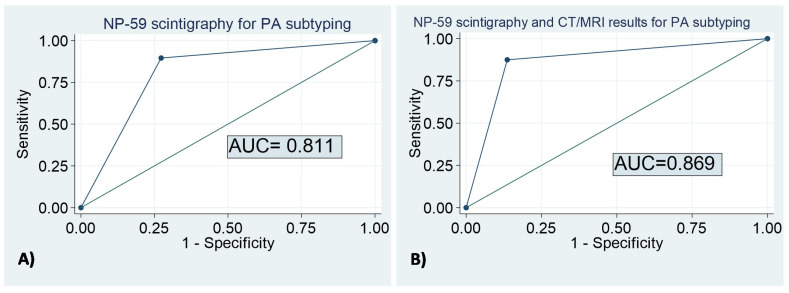
Accuracy of the NP-59 scintigraphy for the subtyping of primary aldosteronism. (**A**) The results of the NP-59 scintigraphy to differentiate between UPA and BPA yielded an area under the ROC curve (AUC) of 0.812 [0.707–0.916]; (**B**) The results of the NP-59 scintigraphy combined to the CT/MRI results to differentiate between UPA and BPA yielded an AUC) of 0.869 [0.782–0.957].

**Table 1 biomedicines-11-01934-t001:** Differences in baseline characteristics between unilateral and bilateral PA cases.

	Unilateral PA (n = 61)	Bilateral PA (n = 25)	*p*
Age (years)	53.3 ± 10.92	56.1 ± 13.66	0.316
Male sex	54.1% (n = 33)	60.0% (n = 15)	0.617
Number antihypertensives	2.6 ± 1.36	1.9 ± 1.20	0.024
Hypertension grade ≥ 2 (n = 75)	76.4% (n = 42/55)	65.0% (n = 13/20)	0.325
Hypertension duration (years) (n = 83)	9.7 ± 7.90	9.5 ± 9.22	0.950
Type 2 diabetes	18.0% (n = 11)	8.0% (n = 2)	0.238
Dyslipidemia	39.3% (n = 24)	56.0% (n = 14)	0.158
Cardiovascular events	29.5% (n = 18)	28.0% (n = 7)	0.889
Cerebrovascular events	6.6% (n = 4)	8.0% (n = 2)	0.812
Chronic kidney disease	8.2% (n = 5)	16.0% (n = 4)	0.283
Sleep apnea syndrome (n = 83)	9.8% (n = 6/61)	4.5% (n = 1/22)	0.444
Active smoking (n = 84)	20.0% (n = 12/60)	25.0% (n = 6/24)	0.614
Hypokalemia at any time	60.7% (n = 37)	56.5% (n = 13)	0.731
Obesity (BMI > 30)	41.0% (n = 25)	36.0% (n = 9)	0.668
BMI (kg/m2) (n = 296)	29.1 ± 5.98	28.5 ± 4.42	0.650
SBP at diagnosis	149.4 ± 22.53	143.2 ± 16.88	0.227
DBP at diagnosis	90.0 ± 12.97	84.1 ± 12.97	0.059
Serum potassium (mEq/L)	3.6 ± 0.55	3.8 ± 0.55	0.113
PAC (ng/dL)	32.8 [22.7–55.1]	36.2 [18.3–52.3]	0.569
PRA (ng/mL/h) (n = 214)	0.2 [0.2–0.34]	0.3 [0.3–1.0]	0.120
PRC (µU/mL) (n = 128)	2.4 [0.9–3.0]	3.0 [1.6–4.6]	0.105
DST > 1.8 µg/dL (n = 30)	62.5% (n = 5/8)	43.8% (n = 7/16)	0.386
Tumor size in CT/MRI of the largest adrenal nodule (mm)	19.1 ± 7.42	18.1 ± 10.51	0.6402
Unilaterality according to CT/MRI *	93.3% (n = 56/60)	40.9% (n = 9/22)	<0.001 †

BMI: body mass index; DBP: diastolic blood pressure; DST: dexamethasone suppression test; PA: primary aldosteronism; PAC: plasma aldosterone concentration; PRA: plasma renin activity; PRC: plasma renin concentration; SBP: systolic blood pressure. † refers to statistically significant results. * Unilaterality, according to CT/MRI, was defined as the presence of a unilateral adrenal nodule > 1 cm and a normal contralateral adrenal gland.

**Table 2 biomedicines-11-01934-t002:** Accuracy of the NP-59 scintigraphy for the subtyping of primary aldosteronism.

	UPA	BPA	Total
Unilateral based on NP-59 scintigraphy	43 (89.6%)	6 (27.3%)	49
Bilateral based on NP-59 scintigraphy	5 (10.4%)	16 (72.7%)	21
TOTAL	48 (100%)	22 (100%)	70

BPA: bilateral primary aldosteronism; UPA: unilateral primary aldosteronism.

**Table 3 biomedicines-11-01934-t003:** Accuracy of the NP-59 scintigraphy results associated with CT/MRI results for the subtyping of primary aldosteronism.

	UPA	BPA	Total
Unilateral based on NP-59 scintigraphy+ CT/MRI	42 (93.3%)	6 (24%)	48
Bilateral based on NP-59 scintigraphy+ CT/MRI	3 (6.7%)	19 (76%)	22
TOTAL	45 (100%)	25 (100%)	70

BPA: bilateral primary aldosteronism; UPA: unilateral primary aldosteronism.

**Table 4 biomedicines-11-01934-t004:** Clinical and hormonal characteristics of patients wrongly classified based on NP-59 scintigraphy results compared to those cases properly classified.

	Uncorrectly Classified (n = 11)	Properly Classified (n = 59)	*p*
Age (years)	56.6 ± 12.03	54.0 ± 11.82	0.498
Male sex	72.7% (n = 8)	45.8% (n = 27)	0.101
Hypertension grade ≥ 2 (n = 75)	90% (n = 9/10)	71.2% (n = 37/52)	0.212
Hypertension duration (years) (n = 69)	14.1 ± 9.48	8.9 ± 8.21	0.064
Hypokalemia at any time	36.4% (n = 4)	59.7% (n = 34)	0.154
Obesity (BMI > 30)	27.3% (n = 3)	44.1% (n = 26)	0.299
Serum potassium (mEq/L)	3.8 ± 0.59	3.6 ± 0.59	0.381
PAC (ng/dL)	27.7 [12.8–107.4]	36.3 [11.9–114.5]	0.339
PRA (ng/mL/h) (n = 214)	0.3 [0.05–0.94]	0.3 [0.05–2.1]	0.250
DST > 1.8 µg/dL (n = 24)	40.0% (n = 2/5)	52.6% (n = 10/19)	0.615
Tumor size in CT/MRI of the largest adrenal nodule (mm)	22.3 ± 6.96	19.9 ± 8.28	0.377
Unilaterality according to CT/MRI (n = 66) *	50.0% (n = 5/10)	83.9% (n = 47/56)	0.016 †

DST: dexamethasone suppression test; PAC: plasma aldosterone concentration; PRA: plasma renin activity; † refers to statistically significant results; * Unilaterality, according to CT/MRI, was defined as the presence of a unilateral adrenal nodule > 1 cm and a normal contralateral adrenal gland.

## Data Availability

The raw data supporting the conclusions of this article will be made available by the authors without undue reservation.

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
