# Peer review of "Diagnostic Accuracy of Adrenal Iodine-131 6-Beta-Iodomethyl-19-Norcholesterol Scintigraphy for the Subtyping of Primary Aldosteronism"

_biomedicines, 2023, doi:10.3390/biomedicines11071934_

Round 1
Reviewer 1 Report
Major revision
Comments to the author:
The authors gave data on unilaterality according to CT/MRI in Table 1, but does this indicate the presence or absence of nodular lesions? If so, please provide a definition (e.g. >1 cm).
Please extract cases without bilateral nodal lesions and compare them with respect to the NP-59 results. Are there cases where there is an accumulation of NP-59 even when nodal lesions are not evident and could be determined to be unilateral?
In conducting this study, please discuss the existence of bias in the use of the NP-59 in selecting cases, such as limiting it to cases where it is clinically difficult to determine whether the disease is unilateral or not.
In general, cases of unilateral PA should tend to have a higher PAC than those with BPA.
What do authors think are the reasons why there is no difference in PAC between UPA and BPA in baseline characteristics? Was there a clinical decision that BPA with mild disease should not be treated with AVS?
Scintigraphy is taken after dexamethasone administration, but is it possible that NP-59 is more detectable in lesions with autonomous cortisol secretion in addition to aldosterone? Please add the LDDST results to the data for the PA cases tested in this study.
Table 2 summarized the accuracy of NP-59. Do the different dexamethasone suppression protocols have an impact on false-negative and false-positive results?
In Figure 2, the accuracy of NP-59 is analysed with a ROC curve. There is only one point, but can the presence or absence of NP-59 accumulation be evaluated as a continuous variable? If there is a numerical value representing the degree of accumulation, it is possible to set multiple diagnostic criteria as a continuous variable.
Author Response
Dear reviewer,
Thank you for your comment and for giving us the opportunity to review our article.
You can find in-depth responses to your comments, beloved. To make it easier to locate modifications made in the updated version of the manuscript, every change has been highlighted in red.
-The authors gave data on unilaterality according to CT/MRI in Table 1, but does this indicate the presence or absence of nodular lesions? If so, please provide a definition (e.g. >1 cm).
Considering the reviewer´s comment we have included the definition of unilaterality based on CT/MRI in the table 1 foot: “Unilaterality according to CT/MRI was defined as the presence of a unilateral adrenal nodule >1 cm and normal contralateral adrenal gland”
-Please extract cases without bilateral nodal lesions and compare them with respect to the NP-59 results. Are there cases where there is an accumulation of NP-59 even when nodal lesions are not evident and could be determined to be unilateral?
Considering the reviewer´s comment, we have included in the results section the results of the proposed analysis:
“When we only included in the analysis, those patients with unilateral PA according to CT/MRI (unilateral ad-renal lesion >1 cm and normal contralateral adrenal gland, n=65), there were 13 cases without uptake in the NP-59 scintigraphy. In the 52 cases in whom hyper uptake was demonstrated, 86.5% (n=45) had unilateral hyper and 13.5% (n=7) bilateral hyper uptake.”
-In conducting this study, please discuss the existence of bias in the use of the NP-59 in selecting cases, such as limiting it to cases where it is clinically difficult to determine whether the disease is unilateral or not.
Considering the reviewer´s comment, we have included a discussion about this point in the limitation section:
“Another point to consider when NP-59 scintigraphy results are evaluated is the potential existence of bias in the use of the NP-59 in selecting cases, such as limiting it to cases where it is clinically difficult to determine whether the disease is unilateral or not. Thus, representing the most challenging cases for the PA subtyping.”
-In general, cases of unilateral PA should tend to have a higher PAC than those with BPA.
What do authors think are the reasons why there is no difference in PAC between UPA and BPA in baseline characteristics? Was there a clinical decision that BPA with mild disease should not be treated with AVS?
We agree with the reviewer´s. However, we did not find differences in PAC between both groups. Considering the reviewer´s comment, we have included a brief discussion about this point in the discussion section:
“The rate of UPA in our study was of 71%, that is quite higher than the described in other series [21]. However, there are other studies reporting similar rates of UPA [22]. The Kim SH study [22] found that the prevalence of UPA was of 70.7% (aldosterone-producing adenoma or unilateral adrenal hyperplasia), while only 29.3% of the patients had bi-lateral disease. These differences may be related to the inclusion of different studies populations across studies, since, for example, it is reported that the rate in UPA is lower in Japanese than in European centers [23]. The prevalence of UPA is also higher in young patients [24]. In addition, we are aware that a potential selection bias may have been occurred in our study since those patients with more chances of being unilateral were more commonly submitted to the AVS, and the rate of hypokalemia and severe hypertension reached prevalences above 60%. Nevertheless, despite this observation, we did not find differences in PAC levels between unilateral and bilateral cases of PA. This finding may be related to the fact that patients with high suspicion of bilateral disease based on CT/MRI and mild PA were less likely to be submitted to AVS and NP-59 scintigraphy and were commonly medically treated without further investigations”
-Scintigraphy is taken after dexamethasone administration, but is it possible that NP-59 is more detectable in lesions with autonomous cortisol secretion in addition to aldosterone? Please add the LDDST results to the data for the PA cases tested in this study.
Thank you for your comment. Considering the reviewer´s comment, we have included the number of patients with associated autonomous cortisol secretion (ACS) in table 1. As you can see, there were few cases of PA tested for ACS (n=30), and no differences in its prevalence were observed between unilateral and bilateral cases of PA. We have also included the prevalence of ACS in the result section:
“The prevalence of autonomous cortisol secretion based on the 1.8 µg/dL threshold in the dexamethasone suppression test was of 50% (n=15/30).”
Table 2 summarized the accuracy of NP-59. Do the different dexamethasone suppression protocols have an impact on false-negative and false-positive results?
Considering the low number of studies performed with the protocol of 4 mg daily, using 1 mg every 6 hours) 7 days before and 2 days after the intravenous injection of NP-59 (n=12), it was not feasible to analyze if there were differences between both protocols. Nevertheless, considering the reviewer´s comment, we have clarified this point in the methods section:
“To suppress the normal adrenal uptake of NP-59, different protocols were used across the different centres: i) the patients took oral dexamethasone 2 mg daily (1.0 mg every 12 h) 3 days before and 7 days after the intravenous injection of NP-59 (n=58); ii) some centers that used a higher dose of dexamethasone ( 4 mg daily, using 1 mg every 6 hours) 7 days before and 2 days after the intravenous injection of NP-59 (n=12).”
In Figure 2, the accuracy of NP-59 is analysed with a ROC curve. There is only one point, but can the presence or absence of NP-59 accumulation be evaluated as a continuous variable? If there is a numerical value representing the degree of accumulation, it is possible to set multiple diagnostic criteria as a continuous variable.
Thank you for your comment. We agree that it would be of interest to have information of the NP-59 accumulation as a continuous variable. However, we do not have this information in the medical records of the patients evaluated.

Reviewer 2 Report
An excellent co-operative and good number study!
Good design (UPA defined by venous sampling and biochemical cure after operation) and analysis for this issue. The results and conclusions are solid, but more information should be offered in order to advance the valuable clinical practice:
Make a detail table about the pathology focusing on false positive and false negative cases of Np59, also discuss the reasons and mechanism.
Correct the mathematic error in table 3
Author Response
Dear reviewer,
Thank you for your comment and for giving us the opportunity to review our article.
You can find in-depth responses to your comments, beloved. To make it easier to locate modifications made in the updated version of the manuscript, every change has been highlighted in red.
-An excellent co-operative and good number study!
Thank you for your overall positive feedback
-Good design (UPA defined by venous sampling and biochemical cure after operation) and analysis for this issue. The results and conclusions are solid, but more information should be offered in order to advance the valuable clinical practice:
Thank you for your comments. We have included the detailed answers below.
-Make a detail table about the pathology focusing on false positive and false negative cases of Np59, also discuss the reasons and mechanism.
Thank you for your comment. Considering the reviewer´s comment we have included a new table with the clinical and radiological information of those cases with false positive and false negative results in the NP-59 scintigraphy. We have compared the clinical and hormonal data of these cases with the data of patients properly classified based on NP-59 scintigraphy results. More over we have included a brief description comparing patients with congruent results in NP-59 scintigraphy and those with incongruent results
“In Table 4 we described the clinical and radiological data of those PA cases wrongly classified as unilateral or bilateral PA according to the NP-59 scintigraphy results compared to those cases properly classified (Table 4). When we compared patients with UPA classified as bilateral PA according to NP-59 scintigraphy and those properly classified as unilateral PA, patients properly classified had unilateral adrenal lesions on CT/MRI more commonly than the former (93.3% vs. 40.9%, P<0.001). However, no differences in tumor size were observed (20.6±7.80 vs. 18±4.24 mm, P=0.290). On the other hand, patients with bilateral PA with congruent NP-59 scintigraphy results had smaller adrenal lesions than those with unilateral uptake (17.5±9.90 vs. 27.6±6.05 mm, P=0.040) and had unilateral adrenal lesions in CT/MRI less commonly (6.7% vs 59.1%, P<0.001).”
-Correct the mathematic error in table 3
Thank you for your comment. We have corrected information of Table 3 as correspond.

Round 2
Reviewer 1 Report
In Table 4, Unilaterality according to CT/MRI of "properly classified" shows n =60. But all number is 59 and it is clearly wrong.
Author Response
Dear reviewer,
Thank you for your comment. We have checked the statistical analysis of the data obtained in Table 4 and we have corrected the data of the column Properly classified (n=59) as correspond.
Table 4. Clinical and hormonal characteristics of patients wrongly classified based on NP-59 scintigraphy results compared to those cases properly classified.
|
Uncorrectly classified (n=11) |
Properly classified (n=59) |
p |
Age (years) |
56.6±12.03 |
54.0±11.82 |
0.498 |
Male sex |
72.7% (n=8) |
45.8% (n=27) |
0.101 |
Hypertension grade ≥2 (n=75) |
90% (n=9/10) |
71.2% (n=37/52) |
0.212 |
Hypertension duration (years) (n=69) |
14.1±9.48 |
8.9±8.21 |
0.064 |
Hypokalemia at any time |
36.4% (n=4) |
59.7% (n=34) |
0.154 |
Obesity (BMI >30) |
27.3% (n=3) |
44.1% (n=26) |
0.299 |
Serum potassium (mEq/L) |
3.8±0.59 |
3.6±0.59 |
0.381 |
PAC (ng/dL) |
27.7 [12.8-107.4] |
36.3 [11.9-114.5] |
0.339 |
PRA (ng/mL/h) (n=214) |
0.3 [0.05-0.94] |
0.3 [0.05-2.1] |
0.250 |
DST > 1.8 µg/dL (n=24) |
40.0% (n=2/5) |
52.6% (n=10/19) |
0.615 |
Tumor size in CT/MRI of the largest adrenal nodule (mm) |
22.3±6.96 |
19.9±8.28 |
0.377 |
Unilaterality according to CT/MRI (n=66)* |
50.0% (n=5/10) |
83.9% (n=47/56) |
0.016† |